# Prognostic Value of Systemic Immune-Inflammation Index (SII) in Patients with Glioblastoma: A Comprehensive Study Based on Meta-Analysis and Retrospective Single-Center Analysis

**DOI:** 10.3390/jcm11247514

**Published:** 2022-12-19

**Authors:** Chao Yang, Bo-Wen Hu, Feng Tang, Qing Zhang, Wei Quan, Jie Wang, Ze-Fen Wang, Yi-Rong Li, Zhi-Qiang Li

**Affiliations:** 1Department of Neurosurgery, Zhongnan Hospital of Wuhan University, Wuhan 430071, China; 2Department of Physiology, School of Basic Medical Sciences, Wuhan University, Wuhan 430071, China; 3Department of Laboratory Medicine, Zhongnan Hospital of Wuhan University, Wuhan 430071, China

**Keywords:** glioblastoma, systemic immune-inflammation index, inflammation, nomogram, prognosis

## Abstract

Inflammation is related to cancer. The systemic immune-inflammation index (SII) has been linked to the prognosis of many types of cancer. The present study aimed to determine the prognostic value of the SII in glioblastoma (GBM) patients based on meta-analysis and single-center retrospective analysis. Relevant publications published before 1 October 2022 were identified by searching PubMed, EMBASE, Cochrane Library databases, and Web of Science. Moreover, 208 GBM patients from Zhongnan Hospital were incorporated. Kaplan–Meier and Cox regression analyses determined the prognostic significance of inflammatory markers. By combining these indicators, we developed scoring systems. Nomograms were also built by incorporating independent variables. The accuracies of nomograms were evaluated by Harrell’s concordance index (c-index) and the calibration curve. According to meta-analysis, an elevated SII predicted the worst overall survival (OS) (Hazard ratio [HR] = 1.87, *p* < 0.001). Furthermore, a higher SII (>510.8) (HR = 1.782, *p* = 0.007) also predicted a poorer outcome in a retrospective cohort. The scoring systems of SII-NLR (neutrophil-to-lymphocyte ratio) showed the best predictive power for OS. The nomogram without MGMT (c-index = 0.843) exhibited a similar accuracy to that with MGMT (c-index = 0.848). A pre-treatment SII is independently associated with OS in GBM. A nomogram integrating the SII-NLR score may facilitate a comprehensive survival evaluation independent of molecular tests in GBM.

## 1. Introduction

Glioblastoma (GBM) is the most frequent kind of malignant central nervous system (CNS) tumor [1]. Despite significant advancements in the treatment of GBM in the past few years, such as maximum safe resection, chemoradiotherapy, and tumor-treating fields, the outcome for patients is still dismal [2,3]. Until now, it has remained difficult to predict the therapeutic response and the prognosis of GBM patients because of the limited realization of valuable biomarkers for GBM. Isocitrate dehydrogenase 1 (IDH 1) and O6-methylguanine-DNA methyltransferase (MGMT) were first introduced to the classification and evaluation of prognosis in patients with GBM based on the 2016 revised World Health Organization (WHO) classification of CNS tumors, which deepened our understanding of genome biology in glioma. Its prognostic value for glioma patients has also been shown [4]. However, the disadvantages of defective technology have limited its widespread application. Additionally, these molecular markers can only be obtained postoperatively. Therefore, it is necessary to identify convenient and effective biomarkers, especially preoperative biomarkers for GBM patients, which may help doctors and patients to make clinical decisions.

It has been known for a long time that there is a biological connection between chronic inflammation and cancer risk [5]. Interactions between tumor cells and inflammatory cells either around the tumor or in peripheral blood can result in a supportive or suppressive microenvironment for tumors. This has aroused the interest of many researchers, and numerous studies have been carried out in recent years [6,7,8]. The hematological peripheral inflammatory markers, easily obtained from routine blood tests, have been investigated as a prognostic factor in several malignant tumors [9,10,11]. Neutrophils, lymphocytes, and platelets are the most common cell types in peripheral blood, and their levels are associated with the body’s immune inflammation status. Due to an increased granulocyte colony-stimulating factor (G-CSF) synthesis from tumor cells, patients with cancers, including glioma, frequently experienced severe neutrophilia and lymphopenia. G-CSF had the potential to shift the lymphocyte lineage to the granulocyte lineage in bone marrow hematopoiesis [12]. The prognostic significance of the neutrophil-to-lymphocyte ratio (NLR) [13] and the lymphocyte-to-monocyte ratio (LMR) [14] has been identified in various types of cancers. Our previous study also showed that the high NLR and low LMR were associated with poor overall survival in GBM patients [15]. Moreover, the significant role of platelets in cancer initiation and development has been explored for a long time [16]. Therefore, an integrated indication based on circulating lymphocyte, neutrophil, and platelet counts may better reflect the inflammatory and immunological condition of the host. The systemic immune-inflammation index (SII) [17], a novel inflammatory biomarker calculated by platelet count*neutrophil count/lymphocyte count, has been determined to be related to worse outcomes in many cancer patients, including breast cancer [18], hepatocellular carcinoma [19] and glioma [20]. Two recent studies reported that a higher SII was associated with a worse outcome in the GBM cohort [21,22]. However, the prognostic value of the SII is still controversial. Yilmaz et al. [23] reported that the SII could not be validated as an independent variable to predict survival in patients with GBM.

The present study aimed to comprehensively explore the prognostic value of the SII in patients with GBM by a meta-analysis and a retrospective study from the single-center cohort. Based on the combination of several prognostic indicators, nomogram models were constructed to predict survival probability for each patient.

## 2. Materials and Methods

### 2.1. Procedures of Meta-Analysis

The meta-analysis was preregistered (12 April 2022) on the International Platform of Registered Systematic Review and Meta-analysis Protocols (INPLASY202240072). The study was carried out following the recommendations in the Preferred Reporting Items for Systematic Reviews and Meta-Analyses (PRISMA) guidelines [24]. The completed PRISMA 2009 Checklist is displayed in Appendix A.

### 2.2. Search Strategy

We searched PubMed, EMBASE, the Cochrane Library database, and the Web of Science for articles that investigated the relationship between the SII and survival in patients with GBM. The publications were published before 1 October 2022. The Mesh Terms included in the meta-analysis were: “Glioblastoma” and “systemic immune-inflammation index”. Other free terms are shown in Appendix A. Reviews and references from included studies were also searched to prevent the omission of pertinent papers.

### 2.3. Eligibility Criteria

Studies were included if they met the following criteria: (1) patients were pathologically diagnosed with GBM; (2) either a preoperative or postoperative peripheral blood test was performed; and (3) the hazard ratio (HR) and 95% confidence interval (CI) for survival were directly accessible via Cox univariate or multivariate analysis in the article. The exclusion criteria were: (1) studies not on humans; (2) not published in English.

### 2.4. Study Selection and Data Extraction

Two authors (C.Y. and B.-W. H.) separately examined the relevant literature and discussed any disagreements. The included studies were read full-text to extract original data. For analyses evaluating the association between the SII and prognosis in GBM patients, the primary information was the HR and 95% CIs. We also extracted the characteristics of the studies, including the first author, year of publication, the number of patients, country, age, sampling time, cut-off value, and type of analysis (univariate analysis [UVA]/multivariate analysis [MVA]).

### 2.5. Quality Evaluation

Independent assessments were made by C.Y. and B.-W. H. of the quality of the studies that were included. The Newcastle–Ottawa Scale (NOS) was utilized to evaluate the quality and risk of bias of the included studies investigating the prognostic usefulness of the SII in GBM patients. Three items, including study selection (0–4 scores), comparability (0–2 scores), and study outcomes (0–3 scores), were used in the process of evaluation. High scores indicated a high quality and low risk of bias.

### 2.6. Inclusion and Exclusion Criteria for GBM Patients

From January 2016 to June 2021, we incorporated 208 adult GBM patients at Zhongnan Hospital of Wuhan University. Inclusion criteria were as follows: (1) age at diagnosis was over 18 years; (2) all the diagnoses were pathologically confirmed based on 2016 WHO classification of CNS tumors; (3) the status of IDH 1 was wild type and (4) data of clinical characteristics, and preoperative peripheral blood routine examination could be obtained. Exclusion criteria were: (1) patients receiving preoperative radiotherapy and/or chemotherapy (including corticosteroids); (2) patients who have had other malignant tumors in the past; (3) recurrent GBM; or (4) patients enduring perioperative death. All procedures followed the Helsinki Declaration [25]. The end date of the follow-up was September 30, 2022. The STROBE checklist was also performed (Appendix A).

### 2.7. Ethics Approval

The present study gained approval from the Ethics Committee of Zhongnan Hospital of Wuhan University (No. 2019048).

### 2.8. Data Collection

We collected data concerning demographic, clinical features, and outcome, including sex, age, locations of tumor, the characteristics of MGMT promoter methylation, Karnofsky performance status (KPS), the extent of tumor resection [subtotal resection (STR) < 100%, gross total resection (GTR) = 100%], and complete postoperative Stupp chemoradiotherapy regimen. Preoperative peripheral blood data of white blood cell (WBC), neutrophil, lymphocyte, platelet, and monocyte count were also reviewed. Using these data, the SII (neutrophil count*platelet count/lymphocyte count), NLR (neutrophil count/lymphocyte count), PLR (platelet-lymphocyte ratio, platelet count/lymphocyte count), and LMR (lymphocyte count/monocyte count) were calculated. The time interval from the operation to death due to any reason or the ending of the last follow-up was utilized to calculate overall survival (OS). The follow-up was performed either in an outpatient setting or by telephone.

### 2.9. Statistical Analysis

The normal distributions of continuous variables were expressed through mean ± standard deviation, and the non-normal distribution is described as the median and interquartile range (IQR) and analyzed by nonparametric tests. Category variables are exhibited as frequency (percentages) and analyzed between groups using chi-square tests. The best cutoffs of inflammatory markers were obtained through X-tile using the minimum *p* value method. (version 3.6.1). Kaplan–Meier plots were generated and examined with the log-rank test (R survminer package) to evaluate the prognostic significance of hematological markers. The independent prognostic significance of these markers was determined using univariate and multivariate Cox regression analyses (R survival package). Variables with independent values were included to establish nomograms to predict the two-year survival rate using the R rms package. The prediction accuracy of the nomogram was tested using Harrell’s concordance index (c-index), and the calibration plot was utilized to analyze the degree to which the values that were predicted and those that were actually observed matched up. All the statistical analysis was performed with R software (version 4.0.2; Institute for Statistics and Mathematics, Vienna, Austria) and SPSS (version 24.0, IBM Corporation, Armonk, NY, USA). Statistical significance was a two-sided *p* value < 0.05. The details of the statistical methods of the meta-analysis are shown in Appendix A.

## 3. Results

### 3.1. A Meta-Analysis Evaluating the Prognosis of SII in Patients with GBM

Ninety-six studies evaluating the relationship of the SII with the survival of GBM patients were initially identified through database searching. Duplicate studies were excluded, and conference abstracts or studies lacking sufficient data were excluded after full-text evaluation. Five studies that enrolled 788 patients were included in the final meta-analysis [20,21,22,23,26]. The selection process of the meta-analysis is shown in Appendix A. The main characteristics of the incorporated studies are shown in Appendix A. A higher SII predicted a lower OS of GBM patients (HR = 1.87, 95% CI: 1.51–2.32, *p* < 0.001) (Figure 1A). The NOS were all seven points, exhibiting a relatively high quality (Appendix A) and the cut-off values differed greatly among them, ranging from 565 to 1200 (10^9^/L). Considering the variables such as sampling time, sample size, and detection method, subgroup analyses were conducted to identify possible sources of heterogeneity (Appendix A). In studies with the preoperative blood sample, the elevated SII was associated with the worst outcome (HR = 1.81, 95% CI: 1.43–2.29, *p* < 0.001), while the SII did not show a significant prognostic value in the subgroup using postoperative samples (HR = 1.92, 95% CI: 0.79–4.67, *p* = 0.150). The SII was also shown to be an important prognostic marker in the present meta-analysis in the subgroups of sample size and detection method. The results were stable according to sensitivity analysis (Appendix A).

### 3.2. Publication Bias

The publication bias of the SII was evaluated using a typical funnel plot, Begg’s funnel plot, and Egger’s linear regression test. (Figure 1B,C). No significant publication bias was detected. (Pr > |z| = 0.806 for Begg’s test and *p* > |t| = 0.968 for Egger’s test).

### 3.3. Clinical Characteristics of GBM Patients from the Single-Center

Basing our research on the meta-analysis, we further assessed the prognostic role of the SII in GBM patients in Zhongnan Hospital retrospectively. The demographic characteristics of all the incorporated patients are shown in Table 1. A total of 208 patients were included in this study, with 84 females and 124 males constituting the cohort. The average age was 57.2 years. Among all the patients, 60.1% of patients had GTR performed on them, and 69.7% of them received standard Stupp chemoradiotherapy postoperatively. The number of patients with MGMT methylation was 89 (42.8%). As shown in Table 1, the patients in the group with a higher SII exhibited a shorter OS (*p* < 0.001).

### 3.4. Prognostic Significance of Peripheral Inflammatory Markers in Patients with GBM

Using X-tile software, we identified the best cutoff of each peripheral indicator. As can be observed in Appendix A, the cutoff values of white blood cell (WBC), neutrophil, lymphocyte, platelet, and monocyte were 6.7, 4.9, 1.9, 255, and 0.4 (10^9^ cells/L), respectively. The cutoff values of NLR, LMR, PLR, and SII were 2.1, 2.3, 249.3, and 510.8 (10^9^ cells/L). Subsequently, the patients were divided into two groups based on the cutoff value of every individual marker. Survival analysis exhibited that a higher WBC (*p* = 0.001), neutrophil (*p* < 0.001), platelet (*p* = 0.018), NLR (*p* < 0.001), PLR (*p* < 0.001) and SII (*p* < 0.001) had a worse outcome, whereas those with a higher lymphocyte (*p* = 0.001) and LMR (*p* < 0.001) had a better OS (Figure 2 and Appendix A). In addition to age, KPS, postoperative standard chemoradiotherapy, the extent of tumor resection, and the status of MGMT, multivariate analysis indicated that the NLR (*p* = 0.017), PLR (*p* < 0.001) and SII (*p* = 0.007) remained independent of prognostic significance (Table 2). Time-dependent ROC analysis demonstrated that the NLR exhibited the best predictive accuracy at one-year and two-year survival rates, followed by the SII and PLR (Appendix A). These results indicated the independent prognostic value of the NLR, SII, and PLR, but not the WBC, neutrophil, lymphocyte, platelet counts, and LMR. Moreover, significant correlations were observed between the SII and NLR (r = 0.898), or PLR (r = 0.776); meanwhile, the NLR was also significantly correlated with the PLR (r = 0.622) (Appendix A).

### 3.5. Prognostic Significance of Scoring Systems in Patients with GBM

Based on the above results, four different scoring systems, such as the SII-NLR, SII-PLR, NLR-PLR and SII-NLR-PLR score, were developed to investigate whether the combinations of these markers can exhibit a more powerful predicting ability. The scores in the three systems were determined by the presence of each marker’s status associated with a worse outcome (NLR > 2.1, PLR > 249.3, SII > 510.8) (Table 3). For the SII-NLR, SII-PLR and NLR-PLR, there were three categories created for the scoring systems: score 0 (neither variable was present), score 1 (any variable was present), and score 2 (both variables were present). The SII-NLR-PLR score was classified into four groups: score 0 (none of these three variables was present), score 1 (any one of the variables was present), score 2 (any two variables were present), and score 3 (all the variables were present). The OS information of each score system is also shown in Table 3, notably, the score 0 and score 2 group of the NLR-PLR system exhibited the longest or shortest survival time among all the score systems. As shown in Figure 3, the SII-NLR, SII-PLR, NLR-PLR and SII-NLR-PLR were all significantly correlated with OS. All three systems including two variables were independent prognostic variables according to multivariate analysis; however, score 1 group in the SII-NLR-PLR system did not exhibit significant independent prognostic value because, possibly, of a relatively small cohort in each group (Table 4). Therefore, the SII-NLR-PLR may be not a good independent variable for OS in GBM. We further assessed the predictive abilities of the SII-NLR, SII-PLR and NLR-PLR score for OS; time-dependent ROC showed that the SII-NLR scoring system was the best variable for predicting OS at one-year and two-year survival rates compared to two other scoring systems and any individual marker (Figure 4 and Appendix A). The details of the cut-off value for the score systems are shown in Appendix A.

### 3.6. Nomograms for Predicting OS in Patients with GBM

Nomogram, a frequent technique used in oncology research as a visual calculating scale model, delivers an estimated digital prognosis for each patient by effectively integrating several prognostic indicators. Therefore, a nomogram incorporating age, the KPS, chemoradiotherapy, the extent of resection, MGMT, and the SII-NLR score was constructed to assess the ability of these variables to predict the two-year survival in patients with GBM (Figure 5A)**.** In the nomogram, the KPS accounted for the most significant percentage, followed by chemoradiotherapy, age, SII-NLR score, resection extent of the tumor, and MGMT. The c-index of the nomogram was 0.848 (95% CI = 0.836–0.861). The bootstrapped calibration plot of the nomogram performed well with the ideal model (Figure 5B). Considering the disadvantages of the molecular test, another nomogram without MGMT was also constructed (Figure 5C). The c-index of this nomogram was 0.843 (95% CI = 0.830–0.855), which was equal to that of the nomogram incorporating MGMT. The bootstrapped calibration plot of the nomogram without MGMT also functioned well with the ideal model (Figure 5D).

## 4. Discussion

Many biological systems and disease processes are closely associated with inflammation, including cancer initiation and progression [6]. The correlation between peripheral inflammation status and the prognosis of cancer patients has been widely reported, including in glioma patients. Among those indexes of peripheral inflammation, the SII, a novel inflammatory biomarker, was shown to be an effective predictor of the outcomes in many solid tumors [27,28,29,30]. In this study, the meta-analysis revealed the clinical significance of a preoperative SII in GBM patients, and this was further verified by the retrospective investigation from single-center data. The SII-NLR score showed the best predictive power for OS compared to the SII-PLR score, the NLR-PLR score, and any other single inflammatory marker. Based on the scoring systems, the nomogram with or without MGMT demonstrated a similar equal predictive accuracy and discrimination for estimating OS.

The research on the relationship between inflammation and a tumor has attracted increasing attention. The systemic inflammatory response is considered as an essential feature of the host response to malignant tumors. It has been found that the systemic inflammatory response can affect the prognosis of tumor patients. Our previous study [15] reported that GBM patients exhibited higher neutrophil counts, NLR, and PLR, as well as lower lymphocyte counts and LMR, compared to those in patients with lower-grade glioma. A high NLR and low LMR were associated with unfavorable OS in GBM patients. Some studies have also indicated that an elevated NLR predicts worse outcomes in patients with colorectal cancer [31], breast cancer [32], and other cancers. The SII, calculated using the absolute measures of platelets, neutrophils, and lymphocytes obtained from routine complete blood counts [19], has been suggested as a biomarker superior to both the NLR and PLR in prognosis prediction [33]. Lei et al. [30] demonstrated that the SII could predict higher pathological grades in young premenopausal endometrial cancer patients. The prognostic value of the SII had also been identified in various types of cancers, including hepatocellular carcinoma [19] for the first time, small-cell lung cancer [10], and glioma [20]. However, a study by Yilmaz et al. [23] reported that the SII could not be used as an independent biomarker for progression-free survival (PFS) and OS in patients with GBM. The meta-analysis of this study, including 788 GBM patients from five individual studies, indicated that elevated SII predicted a poorer OS (HR = 1.87, 95% CI: 1.51–2.32). In the subsequent retrospective study, we also found that a higher value of SII significantly correlated with a worse outcome in patients with GBM (HR = 1.782, 95% CI: 1.168–2.719). However, time-dependent ROC analysis demonstrated that the NLR exhibited the best predictive accuracy at one-year and two-year survival rates, followed by the SII and PLR. Moreover, our analysis showed that the combination of these inflammatory markers displayed more predictive power for OS, compared with any single index. The further result of the nomogram model including the most effective SII-NLR scoring system without MGMT provided the evidence that it is reasonable and practical to evaluate outcomes through simply using the periphery inflammatory indicators and relevant therapy information without molecular tests in patients with GBM.

The precise mechanisms through which the SII affected the prognosis of GBM patients remain unclear. The SII is determined by taking into account the peripheral blood’s neutrophil, platelet, and lymphocyte counts. Evidence suggests that it plays a role in the development, progression, and overall evaluation of many diseases, including human bleeding disorders [34], connective tissue diseases [35], and various malignant tumor forms. Cancers were often featured with neutrophilia, and elevated neutrophils helped the construction of a tumor microenvironment through the secretion of some essential cytokines, such as vascular endothelial growth factor (VEGF) [36], interleukin- 6 (IL-6) [37], IL-8 [38], IL-10 [8], prostaglandin [39], matrix metalloproteinases [40], among others. It benefited the progression, angiogenesis, and distant metastasis of tumors. Some crucial cytokines, including G-CSF, IL-1β, VEGF, and IL-6 [41,42,43] from tumor cells, could potentially increase the number of neutrophils in peripheral blood and tumor tissue. The extent of neutrophil infiltration around the tumor was also reported to be significantly associated with the grade of glioma [44].

The levels of platelets are often elevated in patients with malignant tumors, and the elevated platelets may accelerate the proliferation, angiogenesis, and dissemination of tumor cells through some released factors, such as VEGF and platelet-derived growth factor (PDGF) [45,46,47]. Moreover, the high levels of platelets also had the potential to prevent tumor cells from systemic immune attack and to induce invasive mesenchymal-like phenotype and metastasis by the activation of transforming growth factor β (TGF-β) and nuclear factor kappa-B (NF-κB) pathways [48]. As a result, the enhanced platelet counts were combined with neutrophils to construct a microenvironment beneficial to tumor cell survival. Inversely, lymphocyte was the primary type of immune cells playing an essential role in eliminating tumor cells and immune surveillance in the host [7,49]. Unfortunately, increased neutrophil counts can inhibit lymphocyte survival and normal cytotoxic action by producing reactive oxygen species and arginase [41]. Therefore, the SII was a comprehensive reflection of an impaired immunological function and elevated inflammatory status, instrumental in tumor biological behavior. Furthermore, a study by Liang et al. reported that higher levels of SII were significantly correlated with a higher value of Ki-67, a nuclear protein strongly associated with cellular proliferation, tumor grade, and worse survival in patients with glioma [50].

Interestingly, the study by Wang et al. [10] recently reported that the SII was a valuable tool for doctors to choose therapeutic methods in patients with non-small cell lung cancer (NSCLC). The patients with a higher SII may be suitable for immunotherapy compared to those with a lower SII. Meanwhile, SII dynamics during immunology therapy could also effectively predict the survival of patients with metastatic renal cell carcinoma [51]. Diem et al. [52] analyzed the clinical data of 52 patients with metastatic NSCLC treated with nivolumab. They found that elevated pre-treatment NLR and PLR are associated with shorter OS, PFS, and lower response rates in patients with metastatic NSCLC treated with nivolumab independently of other prognostic factors. Recently, Valero et al. [53] performed a retrospective cohort study of 1714 patients with 16 cancer types treated with immune checkpoint inhibitors (ICI), demonstrating a clinical benefit for a wide range of cancer types. Because only a subset of patients experienced a clinical benefit, there is a strong need for biomarkers that are easily accessible across diverse practice settings. Their results showed that a higher NLR was significantly associated with poorer OS and PFS and lower rates of response and clinical benefit after ICI therapy across multiple cancer types. Further analysis indicated that the NLR could be combined with a tumor mutational burden (TMB) for additional predictive capacity. The probability of benefit from an ICI is significantly higher in the NLR low/TMB high group compared to the NLR high/TMB low group. Moreover, Liu et al. [54] reported that the reduction in the NLR and PLR could predict the clinic remission rate and pathological responsiveness rate, respectively, in patients with rectal cancer. These findings suggest that these peripheral inflammatory indicators may be suitable candidates for a cost-effective and widely accessible biomarker.

There are also some limitations to our study. The number of included studies for meta-analysis was relatively small, and more caution should be applied when interpreting results. The nature of our subsequent retrospective research may lead to a selection bias, and we included a relatively small number of patients in this study. Therefore, prospective and well-designed studies with more samples should be performed to clarify the results. Research on systematic inflammatory biomarkers in cancer treatment decision-making is needed, especially in patients with GBM.

## 5. Conclusions

A pre-treatment SII is a potential prognostic biomarker of OS in patients with GBM. A nomogram that integrates the SII-NLR score may facilitate a comprehensive preoperative survival evaluation independent of molecular tests in patients with GBM.

## Figures and Tables

**Figure 1 jcm-11-07514-f001:**
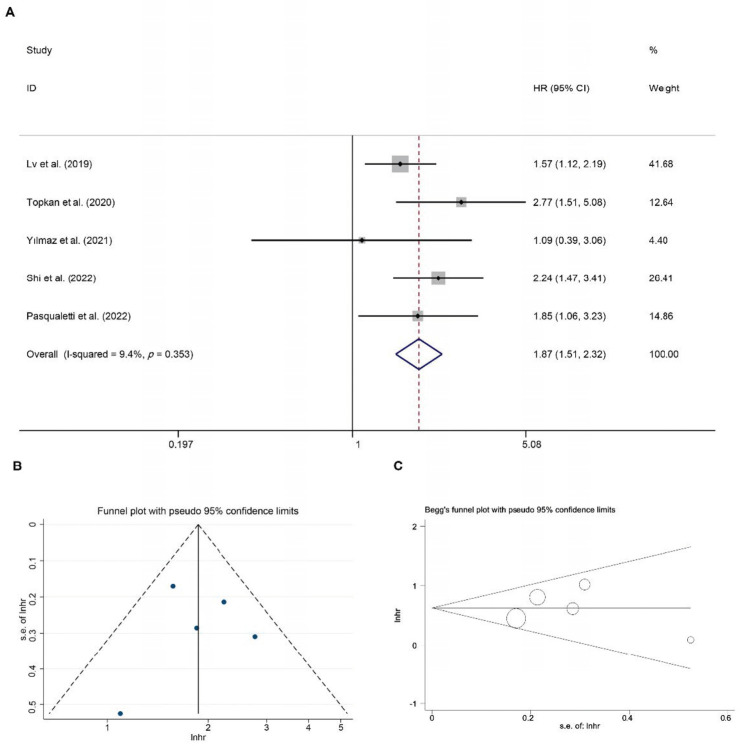
Pooled hazard ratio (HR) and 95% confidence intervals (CIs) of SII for OS in glioblastoma patients (**A**) and detection of publication bias for meta-analysis of survival outcomes based on typical funnel plot (**B**) and Begg’s funnel plot (**C**). The references of Lv et al. (2019), Topkan et al. (2020), Yilmaz et al. (2021), Shi et al. (2022) and Pasqualetti et al. (2022) were [20,21,22,23,26].

**Figure 2 jcm-11-07514-f002:**
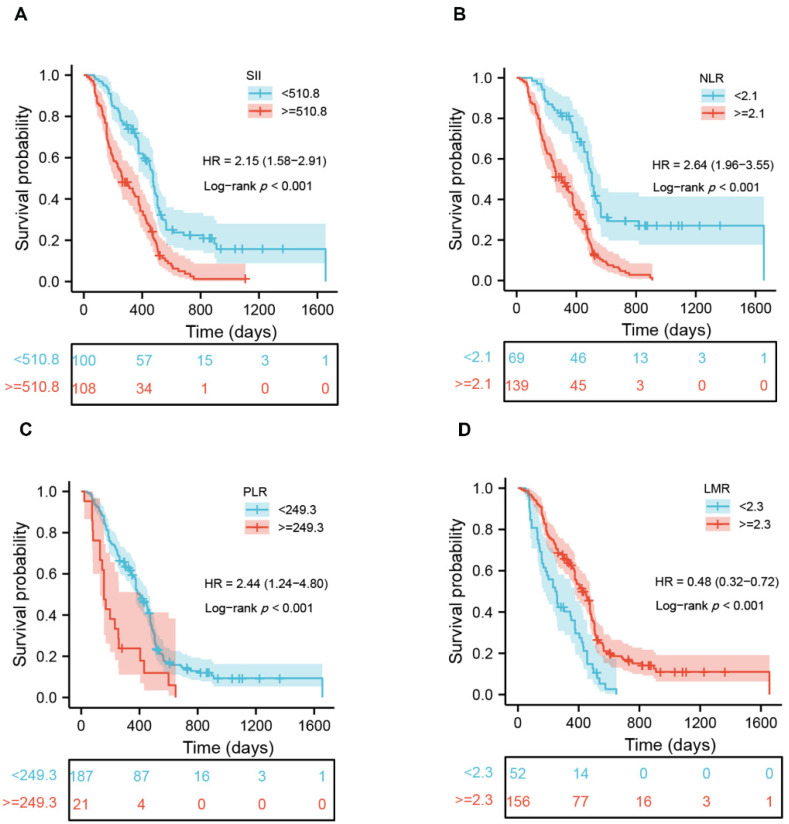
Kaplan–Meier survival curves of glioblastoma patients based on the cutoff values of SII (**A**), NLR (**B**), PLR (**C**), and LMR (**D**).

**Figure 3 jcm-11-07514-f003:**
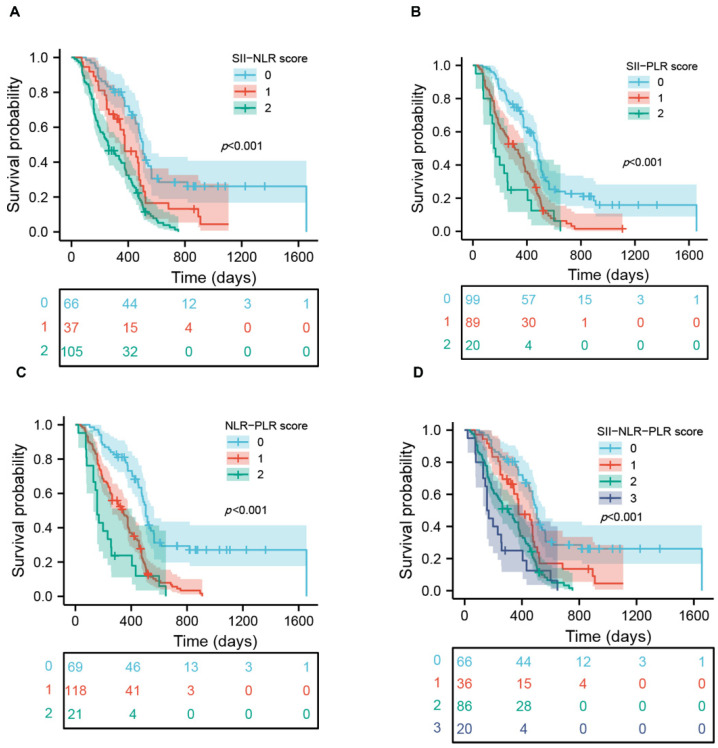
Kaplan–Meier survival curves of glioblastoma patients based on SII-NLR score (**A**), SII-PLR score (**B**), NLR-PLR score (**C**) and SII-NLR-PLR score (**D**).

**Figure 4 jcm-11-07514-f004:**
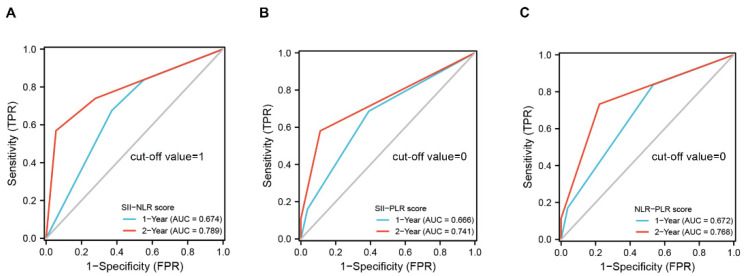
Time-dependent ROC curve analysis of SII-NLR score (**A**), SII-PLR score (**B**), and NLR-PLR score (**C**) for predicting OS at one-year and two-year survival rate in patients with GBM.

**Figure 5 jcm-11-07514-f005:**
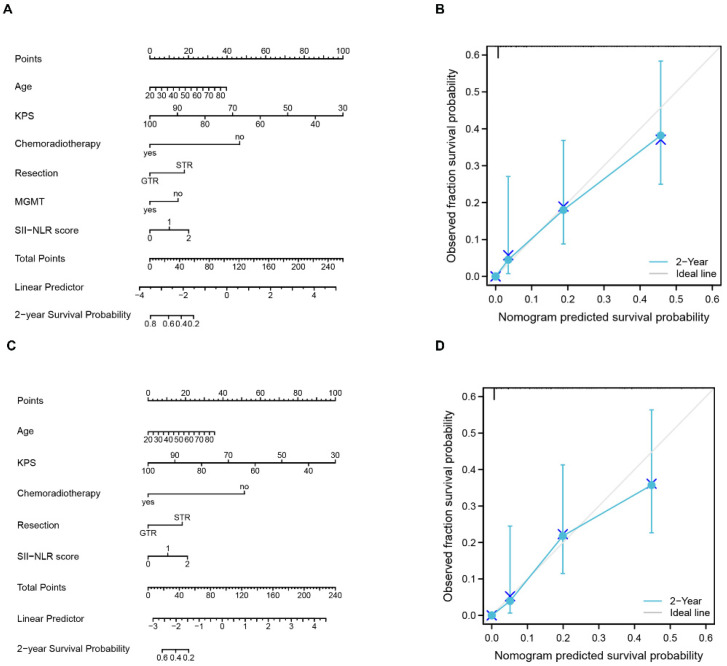
Nomogram and calibration curve. In the nomogram (**A**) and (**C**) ((**A**) the nomogram including MGMT; (**C**) the nomogram without MGMT), each variable was assigned a different score as shown on the largest scale, and the total score for all variables yielded a numerical prediction of two-year survival, with higher scores leading to a worse prognosis. In the calibration curve (**B**) and (**D**) ((**B**) calibration of nomogram A; (**D**) calibration of nomogram C), the grey line represents the ideal prediction, and the blue line represents the performance of the nomogram.

**Table 1 jcm-11-07514-t001:** Clinical and pathological characteristics of patients.

Characteristic	SII < 510.8	SII ≥ 510.8	*p*
*n*	100	108	
Sex, *n* (%)			0.244
female	45 (21.6%)	39 (18.8%)	
male	55 (26.4%)	69 (33.2%)	
Location, *n* (%)			0.667
frontal	32 (15.4%)	31 (14.9%)	
temporal	27 (13%)	26 (12.5%)	
parietal	12 (5.8%)	10 (4.8%)	
multiple	15 (7.2%)	18 (8.7%)	
other	14 (6.7%)	23 (11.1%)	
KPS, *n* (%)			<0.001
≤50	11 (11.0%)	37 (34.2%)	
>50	89 (89.0%)	71 (65.8%)	
Chemoradiotherapy, *n* (%)			<0.001
no	18 (8.7%)	45 (21.6%)	
yes	82 (39.4%)	63 (30.3%)	
Resection, *n* (%)			<0.001
STR	26 (12.5%)	57 (27.4%)	
GTR	74 (35.6%)	51 (24.5%)	
MGMT, *n* (%)			1.000
no	57 (27.4%)	62 (29.8%)	
yes	43 (20.7%)	46 (22.1%)	
Age, median (IQR)	58 (51.75, 65)	59 (51, 65)	0.956
WBC, median (IQR)	5.42 (4.62, 6.39)	7.05 (5.77, 9.17)	<0.001
Neu, median (IQR)	3.28 (2.56, 4)	5.43 (4.11, 7.49)	<0.001
Lym, median (IQR)	1.8 (1.38, 2.13)	1.3 (0.92, 1.62)	<0.001
PLT, median (IQR)	178.5 (143.75, 203.5)	219 (177.5, 273.5)	<0.001
OS, median (IQR)	14.95 (9.95, 18.18)	8.8 (5.2, 14.9)	<0.001

Abbreviations: KPS, Karnofsky performance status; GTR, gross total resection; STR, subtotal resection; IDH, isocitrate dehydrogenase; MGMT, O^6^-methylguanine-DNA methyltransferase; Neu, neutrophil; Lym, lymphocyte; PLT, platelet; OS, overall survival; SII, systemic immune-inflammation index.

**Table 2 jcm-11-07514-t002:** Univariate and multivariate analyses of OS in GBM cohorts.

Characteristics	Total (N)	Univariate Analysis		Multivariate Analysis	
		Hazard Ratio (95% CI)	*p* Value	Hazard Ratio (95% CI)	*p* Value
Sex	208				
male	124	Reference			
female	84	1.032 (0.761–1.400)	0.839		
Age	208	1.025 (1.011–1.040)	<0.001	1.025 (1.010–1.039)	<0.001
Location	208	1.110 (1.004–1.228)	0.042	1.047 (0.944–1.162)	0.385
KPS	208				
≤50	23	Reference			
>50	185	0.150 (0.095–0.237)	<0.001	0.253 (0.148–0.431)	<0.001
Chemoradiotherapy	208				
no	63	Reference			
yes	145	0.066 (0.043–0.103)	<0.001	0.136 (0.075–0.245)	<0.001
Resection	208				
STR	83	Reference			
GTR	125	0.199 (0.145–0.275)	<0.001	0.433 (0.276–0.680)	<0.001
MGMT	208				
no	121	Reference			
yes	87	0.419 (0.306–0.573)	<0.001	0.654 (0.452–0.945)	0.024
WBC	208				
≤6.7	127	Reference			
>6.7	81	1.622 (1.198–2.197)	0.002	0.979 (0.597–1.606)	0.934
Neu	208				
≤4.9	134	Reference			
>4.9	74	2.540 (1.863–3.461)	<0.001	1.040 (0.588–1.839)	0.892
Lym	208				
≤1.9	151	Reference			
>1.9	57	0.546 (0.382–0.782)	<0.001	0.830 (0.550–1.251)	0.373
PLT	208				
≤255	168	Reference			
>255	40	1.546 (1.070–2.234)	0.020	0.710 (0.461–1.096)	0.122
Monocyte	208				
≤0.4	77	Reference			
>0.4	131	1.358 (0.990–1.862)	0.058	1.390 (0.945–2.047)	0.095
NLR	208				
≤2.1	69	Reference			
>2.1	139	2.820 (1.992–3.993)	<0.001	1.769 (1.106–2.829)	0.017
PLR	208				
≤249.3	187	Reference			
>249.3	21	2.478 (1.547–3.969)	<0.001	2.598 (1.473–4.581)	<0.001
LMR	208				
≤2.3	52	Reference			
>2.3	156	0.461 (0.328–0.646)	<0.001	0.899 (0.590–1.371)	0.621
SII	208				
≤510.8	100	Reference			
>510.8	108	2.226 (1.638–3.025)	<0.001	1.782 (1.168–2.719)	0.007

Abbreviations: KPS, Karnofsky performance status; GTR, gross total resection; STR, subtotal resection; IDH, isocitrate dehydrogenase; MGMT, O^6^-methylguanine-DNA methyltransferase; WBC, white blood cell; Neu, neutrophil; Lym, lymphocyte; PLT, platelet; SII, systemic immune-inflammation index; HR, hazard ratio; CI, confidence interval.

**Table 3 jcm-11-07514-t003:** Overall survival based on score system in patients with GBM.

Score				N (%)		OS (Mean ± SD)	*p* Value
Variables	Definition					(Months)	
SII-NLR				208 (100)			
Score 0	SII < 510.8 and NLR < 2.1			66 (31.7)		17.5 ± 9.9	reference
Score 1	SII > 510.8 or NLR > 2.1			37 (17.8)		13.4 ± 7.9	0.012
Score 2	SII > 510.8 and NLR > 2.1			105 (50.5)		9.8 ± 5.8	0.013
SII-PLR				208 (100)			
Score 0	SII < 510.8 and PLR < 249.3			99 (47.6)		16.0 ± 9.2	reference
Score 1	SII > 510.8 or PLR > 249.3			89 (42.8)		10.6 ± 6.4	<0.001
Score 2	SII > 510.8 and PLR > 249.3			20 (9.6)		7.7 ± 5.7	<0.001
NLR-PLR				208 (100)			
Score 0	NLR < 2.1 and PLR < 249.3			69 (33.2)		17.6 ± 10.0	reference
Score 1	NLR > 2.1 or PLR > 249.3			118 (56.7)		11.0 ± 6.2	<0.001
Score 2	NLR > 2.1 and PLR > 249.3			21 (10.1)		7.4 ± 5.6	<0.001
SII-NLR-PLR				208 (100)			
Score 0	SII < 510.8 and NLR < 2.1 and PLR < 249.3			66 (31.7)		17.5 ± 9.9	reference
Score 1	SII > 510.8 or NLR > 2.1 or PLR > 249.3			36 (17.3)		13.8 ± 7.8	0.022
Score 2	SII > 510.8 and NLR > 2.1, or SII > 510.8 and PLR > 249.3, or NLR > 2.1 and PLR > 249.3			86 (41.3)		10.2 ± 5.8	<0.001
Score 3	SII > 510.8 and NLR > 2.1 and PLR > 249.3			20 (9.6)		7.6 ± 5.7	<0.001

SII, systemic immune-inflammation index; NLR, neutrophil-to-lymphocyte ratio; PLR, platelet-to-lymphocyte ratio; OS, overall survival; SD, standard deviation.

**Table 4 jcm-11-07514-t004:** Univariate and multivariate analyses of OS in GBM cohorts based on score systems.

Characteristics	Total (N)	Univariate Analysis		Multivariate Analysis	
		Hazard Ratio (95% CI)	*p* Value	Hazard Ratio (95% CI)	*p* Value
SII-NLR	208		<0.001		
Score 0	66	Reference			
Score 1	37	1.806 (1.139–2.865)	0.012	1.679 (1.053–2.678)	0.030
Score 2	105	3.043 (2.111–4.385)	<0.001	2.460 (1.694–3.572)	<0.001
NLR-PLR	208		<0.001		
Score 0	69	Reference			
Score 1	118	2.642 (1.852–3.769)	<0.001	1.844 (1.269–2.680)	0.001
Score 2	21	4.557 (2.665–7.792)	<0.001	3.687 (2.080–6.535)	<0.001
SII-PLR	208		<0.001		
Score 0	99	Reference			
Score 1	89	2.126 (1.542–2.931)	<0.001	1.623 (1.161–2.267)	0.005
Score 2	20	3.332 (1.997–5.561)	<0.001	2.992 (1.740–5.144)	<0.001
SII-NLR-PLR	208		<0.001		
Score 0	66	Reference			
Score 1	36	1.751 (1.099–2.790)	0.018	1.513 (0.939–2.440)	0.089
Score 2	86	2.892 (1.983–4.216)	<0.001	1.961 (1.321–2.910)	<0.001
Score 3	20	4.219 (2.441–7.293)	<0.001	3.543 (1.987–6.317)	<0.001

SII, systemic immune-inflammation index; NLR, neutrophil-to-lymphocyte ratio; PLR, platelet-to-lymphocyte ratio; OS, overall survival; HR, hazard ratio; CI, confidence interval.

## Data Availability

The original dataset analyzed during the current study is available from the corresponding author upon reasonable request.

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
