# Peer review of "Prognostic Value of Systemic Immune-Inflammation Index (SII) in Patients with Glioblastoma: A Comprehensive Study Based on Meta-Analysis and Retrospective Single-Center Analysis"

_jcm, 2022, doi:10.3390/jcm11247514_

Round 1
Reviewer 1 Report
The paper considers a novel prognostic factor for glioblastoma, SII, which is proportional to platelet*neutrophil count / lymphocyte count.
First, there is a relation between SII and NLR (SII~NLR*PLT), so this should be somehow commented and compared. Also, maybe some other combinations of the factors within the formula for SII probably should be evaluated, like NEUT^alpha*PLT^beta/LYMPH^gamma (i.e. not just direct/inverse proportionality to the factors whih support/inhibit tumor formation, but maybe some power laws would separate the patient clusters better). Some results regarding comparison between SII and NLR are visible in Tab. 1: in univariate analysis, NLR performs better than SII, the contrary happens in multivariate analysis.
From the scoring mechanism it follows (Tab. 3), that SII and NLR can work as independent prognostic factors; however to make this more solid it would be good to see the treshold point on the histogram of considered SII/NLR scores.
Considering the scoring mechanism: why only pairwise scoring, SII-NLR, SII-PLR, NLR-PLR, why not joint score for SII+NLR+PLR? Maybe the results would be even better if all of them are independent? Also (Tab. 3), it would be good to mark, that within the scoring framework the best results are displayed by the NLR-PLR pair (the longest survival in the "healthty group", the shortest survival in the weak group).
Reviewer 2 Report
Dear Editor,
Thank you for sending me this article, well written and pleasant to read.
I have just one comment to raise. Considering that not all patients retrospectively evaluated underwent standard treatment, would it be possible to assess both the prognostic and predictive power of the inflammation indices?
